# Fibrous Materials Made of Poly(*ε*-caprolactone)/Poly(ethylene oxide)*-b-*Poly(*ε*-caprolactone) Blends Support Neural Stem Cells Differentiation

**DOI:** 10.3390/polym11101621

**Published:** 2019-10-08

**Authors:** Daniel Fernández, Montserrat Guerra, Judit G. Lisoni, Thomas Hoffmann, Rodrigo Araya-Hermosilla, Toshimichi Shibue, Hiroyuki Nishide, Ignacio Moreno-Villoslada, Mario E. Flores

**Affiliations:** 1Instituto de Ciencias Químicas, Facultad de Ciencias, Universidad Austral de Chile, Valdivia 5090000, Chile; daniel.fernandez01@alumnos.uach.cl (D.F.); imorenovilloslada@uach.cl (I.M.-V.); 2Instituto de Anatomía, Histología y Patología, Facultad de Medicina, Universidad Austral de Chile, Valdivia 5090000, Chile; monserratguerra@uach.cl; 3NM MultiMat, Instituto de Ciencias Físicas y Matemáticas, Facultad de Ciencias, Universidad Austral de Chile, Valdivia 5090000, Chile; judit.lisoni@uach.cl (J.G.L.); thomas.hoffmann@uach.cl (T.H.); 4Programa Institucional de Fomento a la Investigación, Desarrollo e Innovación (PIDi), Universidad Tecnológica Metropolitana, Santiago 8940577, Chile; rodrigo.araya@utem.cl; 5Material Characterization Central Laboratory, School of Science and Engineering, Waseda University, Tokyo 169-8555, Japan; shib@waseda.jp; 6Research Institute for Science and Engineering, Waseda University, Tokyo 165-8555, Japan; nishide@waseda.jp

**Keywords:** biocompatible polymers, amphiphilic block copolymers, semicrystalline polymers, electrospinning, biohybrid materials, stem cells differentiation

## Abstract

In this work, we design and produce micron-sized fiber mats by blending poly(*ε*-caprolactone) (PCL) with small amounts of block copolymers poly(ethylene oxide)_m_*-block-*poly(*ε*-caprolactone)_n_ (PEO_m_-*b*-PCL_n_) using electrospinning. Three different PEO_m_-*b*-PCL_n_ block copolymers, with different molecular weights of PEO and PCL, were synthesized by ring opening polymerization of ε-caprolactone using PEO as initiator and stannous octoate as catalyst. The polymer blends were prepared by homogenous solvent mixing using dichloromethane for further electrospinning procedures. After electrospinning, it was found that the addition to PCL of the different block copolymers produced micron-fibers with smaller width, equal or higher hydrophilicity, lower Young modulus, and rougher surfaces, as compared with micron-fibers obtained only with PCL. Neural stem progenitor cells (NSPC), isolated from rat brains and grown as neurospheres, were cultured on the fibrous materials. Immunofluorescence assays showed that the NSPC are able to survive and even differentiate into astrocytes and neurons on the synthetic fibrous materials without any growth factor and using the fibers as guidance. Disassembling of the cells from the NSPC and acquisition of cell specific molecular markers and morphology progressed faster in the presence of the block copolymers, which suggests the role of the hydrophilic character and porous topology of the fiber mats.

## 1. Introduction

Healing ability is an intrinsic property observed in the human body [1]. However, not always damaged tissues can self-heal under natural mechanisms. In this context, tissue engineering plays a crucial role in providing suitable complex materials that help in the healing process. These materials may be conformed by a scaffold as support of cell growing and tissue formation, bioactive agents, and, optionally, cells. A key topic in tissue engineering is to provide scaffolds with suitable degradation rates and functionality by means of the design of 3D advanced materials, such as supplying interconnected pores for cell-cell and cell-scaffold interactions, and regulating cell phenotype [2,3,4,5]. In order to bring cells together to form a tissue, the cell-scaffold interaction may mimic the interaction with the natural support, the so-called extracellular matrix (ECM) [6]. ECM is a complex and intricate macromolecular network where all tissues and organs are contained as a mixture of cells and non-cellular components. ECM is composed by different macromolecules with different functionality according to the tissue, such as proteins and polysaccharides, that are locally secreted by specific cells [7,8]. Tissue engineering technologies use similar macromolecular networks in order to build a functional tissue composed by cells and an appropriated ECM support. Depending on the tissue, different strategies are followed in the preparation of scaffolds, such as 3D printing, electrospinning, solvent casting, freeze drying, and gas foaming [5,9,10]. For example, bone tissue engineering uses 3D printed scaffolds of regular porosity and shape, which include inorganic nanoparticles as bioactive agents, in order to provide an adequate environment to bone regeneration [11,12,13,14,15]. Highly porous scaffolds with less ordering, based on the ice-segregated ionic self-assembly of complementary charged polysaccharides, have been obtained by freeze-drying for dermis regeneration [16,17].

In the case of the nerve system, damage is a common clinical disease. When the nerve injury gap is too long, the healing mechanism in the human body fails at repairing it [18,19]. In this context, helping nerve tissue regeneration represents a challenge [20,21]. A proper scaffold should provide to the neural cells the adequate environment to grow and rebuild nerve network [22,23,24]. The study in-vitro of neural-active scaffolds may be done using neural stem cells (NSC). NSC are a type of multipotent stem cells with the ability to self-renew, property related to their role in the generation of the nervous system for all mammalian during the embryonic stage [25]. These stem cells present the ability to differentiate into different lineages such as neurons, astrocytes, and oligodendrocytes [26], which is determined by several factors including differentiation and growth factors, cell–cell and cell–matrix interactions. In addition, these cells may serve for creating neural tissues in-vitro [27]. The creation of tridimensional materials containing cells represent a technological advance for applications such as drug screening, which allow studying the behavior of cells in-vitro after applying external stimulus, thus avoiding the use of animals in these studies [28,29].

The use of biocompatible synthetic polymers for the design of synthetic tissues offers some advantages, which may provide materials with similar composition to native ECM [30,31]. Although natural polymers such as gelatin, chitosan, and dextran are broadly used, due their molecular similarity with native ECM components, the use of biocompatible synthetic polymers allows the control in the macromolecular structure, comonomeric composition, molecular-weight and distribution [32]. In principle, by using different polymer blends, it is possible to modulate materials properties to mimic properties close those observed in native ECM. Poly(*ε*-caprolactone) (PCL) is an ideal starting polymer to produce biocompatible and biodegradable synthetic materials. However, the resulting PCL materials are hydrophobic, hindering its application on biological liquid media [33,34]. One key problem in tissue engineering is the production of materials with controllable hydrophilia, because cells under culture are more attracted to partial wetting surfaces than those presenting more hydrophilic or hydrophobic character [35]. Although the tuning of the hydrophobia of PCL materials can be done by different methods such as surface hydrolysis [36,37] and plasma treatment [38,39], blending of hydrophobic polymers with hydrophilic polymers, such as poly(ethylene oxide) (PEO) [40,41,42,43], offers a cost-effective method to modify the physicochemical properties of materials in terms of wettability, mechanical properties, and cell-matrix interactions [44,45]. PCL and PEO are widely used in tissue engineering and drug delivery. However, due to the incompatibility between PCL and PEO, both polymers may undergo crystallization in the solid state in different phases [46]. However, depending on the concentration of the hydrophilic polymer in the blend, confinement of the hydrophilic polymer into the hydrophobic component lamellae is observed [47,48]. In order to minimize phase separation, blending the mixture of both homopolymers with PEO-*b*-PCL block copolymers has proven to be useful [40,41].

Electrospinning is a technique that allows the preparation of non-woven fibrous materials with similar structure to that observed in native ECM [49]. This processing technique uses electrostatic forces in order to stretch a polymeric jet solution into a continuous fiber. Several polymeric solutions can be processed by electrospinning, including homopolymers such as PEO and PCL, as well as corresponding blends [47,50,51]. Blend electrospinning is an efficient strategy to combine the advantages of different polymers in the resulting electrospun materials, allowing improved mechanical and functional properties [52,53]. In order to modify the surface of hydrophobic polymeric electrospun fibers, the use of block copolymers bearing hydrophobic and hydrophilic segments in low concentration offers a different approach to tune the final material properties. The hydrophobic segment of the block copolymer can act as an anchor to the hydrophobic bulk component, meanwhile the hydrophilic block can locate at the surface of the materials [54,55].

In this work, we design and produce micron-sized fiber mats as synthetic ECM by electrospinning. PCL_394_ has been used as main component of the synthetic ECM to which we added small amounts of PEO_m_*-b-*PCL_n_ block copolymers with different degree of polymerization in both blocks. The influence of these block copolymers in the hydrophilic/hydrophobic character of the fiber mats has been evaluated. In addition, size, surface morphology, chemical, and mechanical properties of the materials are characterized by SEM, X-ray diffraction (XRD), small angle X-ray scattering (SAXS), thermomechanical analysis (DMA and DSC), and tensile tests. The ability of neural stem progenitor cells (NSPC) obtained from rat brain to differentiate in the presence of the fabricated synthetic ECM and in the absence of any added growth factor has also been evaluated. This strategy points, then, at mimicking the possible role of ECM in neurogenesis, aiming at the fabrication of tissue substitutes and organoids.

## 2. Materials and Methods

### 2.1. Materials

Polyethylene oxide monomethyl ether (PEO_45_, PEO_148_, PEO_230_, Sigma Aldrich), poly(*ε*-caprolactone) (PCL_394_, average M_n_ 45,000 g/mol, Sigma Aldrich, St. Louis, MO, USA), *ε*-caprolactone (*ε*-CL, purity: 97%, Sigma Aldrich, St. Louis, MO, USA) (see Figure 1), stannous (II) octoate (SnOct_2_, purity: 95%, Sigma Aldrich, St. Louis, MO, USA), tetrahydrofuran (THF, purity ≥99.5%, Merck, Darmstadt, Germany), *n*-hexane (purity ≥98.5%, Merck, Darmstadt, Germany), and dichloromethane (DCM, purity ≥99.8%, Merck, Darmstadt, Germany), were used as received. Culture medium (NeuroCult and NeuroCult NSA proliferation medium-rat, STEMCELL Technologies, Vancouver, BC, Canada) were used to culture stem cells. The following primary antibodies were used for immunofluorescence assays: (i) glial fibrillary acidic protein (GFAP, astrocyte marker), polyclonal raised in rabbit (1:750 dilution, Sigma, St. Louis, MO, USA); (ii) β-III tubulin (neuronal marker), monoclonal (1:750 dilution, Sigma, St. Louis, MO, USA); 1 mg/mL DAPI solution for nucleus staining (4′,6-diamidino-2-phenylindole dihydrochloride, Molecular Probes, Thermo Fisher Scientific, Waltham, MA, USA). Secondary antibodies conjugated with Alexa Fluor 488 or 594 (1:500 dilution, Invitrogen, Carlsbad, CA, USA) were also used.

### 2.2. Equipment

The electrospinning apparatus consists of a luer-lock 5 mL glass syringe (Hamilton Gastight, Franklin, MA, USA) connected to a PTFE tubing with a PEEK luer-lock connector with a flat-end 22G metallic needle, a syringe pump (NewEra Pump Systems Inc, Farmingdale, NY, USA) to control the feeding rates, a grounder copper plate covered with rectangular sheet of aluminum foil, and a high voltage DC power supply (Genvolt High Voltage Ind. Ltd., Bridnorth, United Kingdom) connected to the metal needle and the ground copper plate. SEM images of fibrous materials were obtained in CrossBeam Scanning Electron Microscope (FIB-SEM Auriga Compact, Zeiss, Aalen, Germany). Wettability of fibrous materials was measured by sessile drop method in a contact angle meter (Holmarc HO-IAD-CAM-01). X-ray diffraction was measured in D2 Phaser diffractometer (Bruker, Billerica, MA, USA). Small angle X-ray scattering were done in Laboratório Nacional Luz Síncrotron (Campinas, Brazil). Thermal properties of polymer and fibrous materials were measured using a dynamic mechanical analyzer (DMA 8000, Perkin Elmer, Waltham, MA, USA) and differential scanning calorimeter (DSC 8000, Perkin Elmer, Waltham, MA, USA). Tensile test were carried by stress-strain tests obtained using a texture analyzer (CT3-1000, Brookfield, Middleboro, MA, USA) under tension mode. Immunofluorescence images were obtained in a fluorescence microscope (AxioImager Z2, Zeiss, Aalen, Germany).

### 2.3. Methods

#### 2.3.1. Block Copolymer Synthesis

Block copolymers of PEO_m_-*b*-PCL_n_, with different degree of polymerization of PEO and PCL block, have been synthesized by bulk ring-opening polymerization of *ε*-caprolactone, using stannous (II) octoate as catalyst. PEO_45_, PEO_148_, and PEO_230_ monomethyl ether were used as macroinitiators in order to produce diblock copolymers with different degree of polymerization of PEO [56,57]. Besides, the control in the degree of polymerization of PCL has been achieved by changing the molar ratio of *ε*-caprolactone monomer to PEO macroinitiatior. In a dried Schlenk flask, 2 g (0.2 mmol) of macroinitiator PEO_m_ monomethyl ether, previously dried in a vacuum oven at 50 °C for 24 h, was added. Variable amounts of *ε-*caprolactone were added under nitrogen atmosphere, to achieve different defined monomer to macroinitiator ratio. The mixture was stirred until the dissolution of the PEO macroinitiator was observed. Then, four drops of pure SnOct_2_ was added under nitrogen atmosphere. Three freeze-pump-thaw cycles were applied in order to degasify the mixtures, and, finally, the Schlenk tubes were left in nitrogen atmosphere. The flasks were heated at 135 °C with constant stirring during 6 h. The resulting viscous liquid was cooled to room temperature, dissolved in 10 mL of tetrahydrofuran, and precipitated into a large excess of cold n-hexane. The resulting precipitate was filtered, and redissolved in tetrahydrofuran and reprecipitated in cold n-hexane. The white precipitated was collected by filtration and vacuum dried at 35 °C for one day. The copolymerization reactions proceeded with a high yield, over 90 wt%. Detailed characterization (structure by ^1^H-NMR and molecular weight by GPC) of block copolymers synthesis can be found in the Appendix A).

#### 2.3.2. Fiber Fabrication and Characterization

For electrospinning experiments, we prepared four different solution in DCM as solvent containing 20 wt% of PCL_394_ alone and in the presence of 2.5 wt% of the block copolymers PEO_45_-*b*-PCL_11_, PEO_148_-*b*-PCL_13_, and PEO_230_-*b*-PCL_180_, respectively. After that, the polymeric solutions were loaded in a 5.0 mL glass syringe. The distance between needle and collector was kept at 21 cm. The mats were obtained by depositing 700 μL of polymer solution at a feed rate of 50 μL/min, applying an electric voltage between the needle and collector of 10.0 kV. The environmental relative humidity at the laboratory was controlled and set at 40%. Fibers were collected on aluminum foil and in microscope glass slices for further analysis. SEM measurements were done by sticking a small rectangle of 0.5 cm^2^ of fibrous material on carbon tape mounted into a standard SEM specimen stub, and further sputtered with a thin layer of gold. For wettability, a small rectangle of 2 cm^2^ of fibrous material was cut and stored into a desiccator during three days in order to avoid surface moisture. After that, samples were loaded into the contact angle meter, and a drop of water (50 µL) was deposited in the surface of the fibers. Evolution in the drop shape on the material was recorded in a video at 17 fps. Images captured at the beginning of experiments were labeled as t_0_, meanwhile those representing the last image were represented as *t_f_*. XRD diffractograms were measured using a monochromatic Cu K-α radiation (1.54184 Å). Diffraction patterns were collected at 0.002°/s with 2θ ranging from 10° to 50°. Small angle X-ray scattering (SAXS) measurements were done in the SAXS1 beamline of the Brazilian Synchrotron Light Laboratory (LNLS-CNPEM, Campinas, Brazil). The synchrotron line is operated at 8 keV with a resolution of 0.1 (*ΔE/E*) and a spot size on the irradiated sample of 1.5 × 1.0 mm^2^. Samples were measured at room temperature using 250 s of exposition. Scattering data was collected by a Pilatus 300 K hybrid pixel X-ray detector (Dectris), keeping a sample to detector distance of 3.0 m, which offers a range of scattering angle (q) of 0.04–1.5 nm^−1^. Scattering data was further processed by Python Spyder3 and SaSView (version 4.2.2) software. For DMA measurements, the fibrous materials were loaded into a stainless-steel material pocket with dimensions of 30 x 14 mm^2^, and then the pocket was folded, achieving thickness of around 1 mm. The pocket was clamped in the DMA equipment using the single cantilever geometry. The storage modulus (*E’*), and tan delta (*tan*
*δ*) were measured from −100 to 100 °C, heating at 2 °C/min, applying a static force of 2 N, 1 Hz of frequency, and 0.05 mm of strain. The corresponding glass transition (*T_g_*) were determined from the position of the peaks in the *tan*
*δ* curves. Melting (*T_m_*) and cold crystallization (*T_c_*) of bulk polymers and PCL_394_/PEO_m_-*b*-PCL_n_ fibers were measured by DSC. The samples were weighed (10–17 mg) in an aluminum pan, which was then sealed. Hereafter, the samples were heated from 0 to 100 °C and then cooled to 0 °C. Two heating–cooling cycles were performed at a rate of 10 °C/min in N_2_ atmosphere. Integration of melting peaks were done in order to calculate the enthalpy of melting. DMA and DSC data were analyzed by Pyris Diamond software (Shelton, CT, USA). Tensile test were done cutting the fibrous mats in rectangles of 12.5 cm of length by 6.0 cm width. Samples were tested using 1.0 g of initial load, until 50% of deformation, at a speed of deformation of 0.50 mm/s. The Young modulus of fibrous materials was calculated using the slope of the initial linear zone observed in the stress-strain plot. Due limitations of the equipment used to measure tensile test, we performed measurements up to 270% of deformation, in order to obtain the elongation at break of materials.

#### 2.3.3. Neurospheres from the Brain of Sprague-Dawley (SD) Rats

SD rats were bred, housed, handled and cared in the Animal Facility of the Universidad Austral de Chile, operating under standard protocols established by the National Research Council of Chile (CONICYT), and the Ethics Committee of Universidad Austral de Chile (Project identification code: 0042/18 (Fondecyt Iniciación 11811029); 0065/15 (Fondecyt Postdoctorado 3160648)). 1 day postnatal SD rats were killed by decapitation. The brain was isolated, and the dorsolateral walls ventricular/subventricular zone (VZ/SVZ) were isolated from the brain. The tissue obtained was stored in a sterile 2.0 mL plastic microtube containing 1.0 mL of culture medium prepared in the presence of epidermal growth factor (20 ng/mL), heparin (2 μg/mL), and antibiotics penicillin/streptomycin (100 μg/mL). The cells were isolated applying a mechanical disintegration of the tissue, adding 1.0 mL of culture medium, and centrifuging the mixture of cells and tissue during 10 min at 110 g. The obtained pellet was redissolved in 2.0 mL of culture medium, and the viable cells obtained were quantified by hemocytometry using the trypan blue test. In order to obtain neurospheres, 60,000 cells per mL were seeded in a non-adherent plastic culture plate. The cells were maintained in a culture oven at 36 °C and in a 5% CO_2_ atmosphere during 6 days in vitro (6DIV). The cells were monitored daily through optical microscopy.

#### 2.3.4. Differentiation Assay

Prior to biological assays, fibrous materials were left in a laminar air flow cabin during one week in order to allow evaporation of occluded/residual solvent. The resulting neurospheres (6DIV culture) were collected from grown single neural stem or neural progenitor cells (NSPC), plated on PCL fiber mats and cultured in a sterile plastic culture dish containing culture medium prepared in the absence of epidermal growth factors. The cells were exposed to the fibrous materials during for 7 days, and monitored daily through optical microscopy.

#### 2.3.5. Whole Mount Immunofluorescence

After 7 days in culture, PCL fiber mats and seeded NSPC were isolated from the culture medium. The cells present in this culture were fixed during 15 min in a 2% paraformaldehyde solution (prepared in phosphate buffer (PBS) at pH 7.4), and then postfixed during 15 min using a 4% paraformaldehyde solution (prepared in PBS at pH 7.4). Fixed cells were processed for double immunofluorescence analysis adding both anti-*β*III tubulin and anti-GFAP antibodies, and DAPI marker. The dilution of antibodies was 1:750, using a buffer solution 0.1 M Tris buffer at pH 7.8, prepared 0.7% of non-gelling seaweed gelatin and lambda carrageenan, and 0.5% of Triton X-100. The secondary antibodies, conjugated with Alexa Fluor 488 or 594, in a 1:500 dilution, were used. Incubation of antibodies were carried out during 18 h at room temperature. Fixed cells in the mats were coverslipped by using a mounting medium (Vectashield, Dako, Santa Clara, CA, USA) and inspected under an epifluorescence microscope (AxioImager Z2, Zeiss, Aalen, Germany) in order to study differentiated NSPC by using the multidimensional acquisition software AxioVision Rel version 4.6 (Zeiss, Aalen, Germany).

## 3. Results

### 3.1. PEO_m_-b-PCL_n_ Synthesis and Blend with PCL_394_

By ring opening polymerization (ROP) it is possible to obtain different block copolymers with variable hydrophilic and hydrophobic length segments to prepare mixtures with PCL_394_. The degree of polymerization of PCL into the block copolymer can be controlled by the monomer (*ε*-caprolactone) to initiator (PEO) ratio (M/I). We obtained the block copolymers PEO_45_*-b-*PCL_11_, and PEO_148_*-b-*PCL_13_ using a M/I ratio of 10, and PEO_230_*-b-*PCL_180_ using a M/I ratio of 180. These block copolymers present molecular weights ranging from 3400 to 34,000 g/mol (see Appendix A for detailed characterization). The first two block copolymers have a short PCL block, accompanied of a small and medium size PEO block. The latter presents both blocks of medium size, showing a total polymerization degree almost equal to the PCL homopolymer. PEO_m_*-b-*PCL_n_ and PCL_394_ are polymers with a tendency to solubilize in organic solvents such as dichloromethane (DCM) or tetrahydrofuran (THF), due to the similar Hansen solubility parameters of both polymers and solvents. The selection of a common solvent for both PEO_m_*-b-*PCL_n_ and PCL_394_ can be important to obtain a solid material where homogeneous distribution of the blended components is required [58,59,60]. We choose DCM as solvent because the mixture with PCL_394_ and PEO_m_*-b-*PCL_n_ produces clear and homogeneous solutions.

### 3.2. Scaffold Fabrication

Electrospun mats were obtained by using solutions of PCL_394_ at 20 wt% in DCM in the absence or in the presence of 2.5 wt% of the block copolymers. SEM images in Figure 2 show that all the PCL_394_ mats synthesized present an interconnected structure similar to the ECM [61,62]. Interestingly, PCL_394_ fibers present a bimodal distribution of the mean fiber diameter centered at 3 and 10 μm as frequency histograms in Figure 2 show. It can be clearly seen that the presence of any of the block copolymers reduces the mean diameter of the fibers, ranging between 0.2 and 4 μm. Fibers containing PEO_230_*-b-*PCL_184_ were the less polydisperse. In Figure 2, it can be also observed that the presence of the block copolymers induces surface irregularities in the resulting fibers. Pristine PCL_394_ fibers present a relatively flat surface. By blending PCL_394_ with PEO_45_*-b-*PCL_11_, porous topologies appear on the surface of the fibers. The effect is enhanced when blending with the other two copolymers containing a higher degree of PEO polymerization that produces fibers with more wrinkled surfaces. These surface irregularities appear as a consequence of both phase separation of the polymers in the fibers [63] and an increase on the water condensed during electrospinning process due to the PEO block hydrophilia [47,64,65,66].

### 3.3. Fiber Mat Characterization

#### 3.3.1. Hydrophilic/Hydrophobic Character of the Fiber Mats 

In Figure 3, it is shown the images of a water drop in contact with the fibrous materials. As previously mentioned, PCL is a highly hydrophobic polymer, which is reflected in the high initial contact angle (121.0 ± 8.2° at *t_0_*) in PCL_394_ fibers. The addition of block copolymers showing low PCL degree of polymerization, PEO_45_*-b-*PCL_11_ and PEO_148_*-b-*PCL_13_, produces highly hydrophilic fibers, revealed by the lower initial contact angle (59.0 ± 7.2° and 40.3 ± 9.2° at *t_0_*, respectively). Interestingly, the fibrous materials prepared with the block copolymer with similar and medium degree of polymerization of both PEO and PCL blocks (PEO_230_*-b-*PCL_184_) results in a hydrophobic mat (124.8 ± 8.4° at *t_0_*). At a certain time after deposition of the water drop, mats of PCL_394_ remain hydrophobic (117.1 ± 17.7° at *t_f_* = 5 min). The same trend is observed with mats prepared by blending PCL_394_ with the block copolymer PEO_230_-*b*-PCL_184_ (117.6 ± 8.4° at *t_f_* = 20 min). In mats containing the block copolymers PEO_45_-*b*-PCL_11_ and PEO_148_-*b*-PCL_13_ the water drop collapsed in less than of 1 s, and the material got completely wet.

The key point of these facts stems from the dual hydrophilic and hydrophobic behavior of PEO block. When the polymerization degree of PCL in the copolymers increases in the block copolymer, crystallization of the PEO block is suppressed whereas crystallization of the PCL block is dominant [67]. On the contrary, when a short PCL block is present in the block copolymers, such as in PEO_45_*-b-*PCL_11_ and PEO_148_*-b-*PCL_13_, crystallization of the PEO block at the surface of the fibers may furnish a higher hydrophilic character [40,68]. It can be suggested that the copolymer architecture is a good tool to tune the hydrophilic/hydrophobic character of the mats.

#### 3.3.2. Internal Structure of the Fibers

Macromolecules of PCL in the fibers crystalize oriented along the fiber axis in the form of lamellae [69]. Crystals are intercalated with amorphous domains in an alternating pattern. Some characteristic lengths of the internal structure of the fibers are given in Table 1, related with the cartoon shown in Figure 4, attributed to PCL_394_ as major component. Characteristic lengths corresponding to planes (110) and (200) could be found by XRD measuring full width at half maximum for the corresponding diffraction peak and applying the Scherrer equation (see Appendix A for detailed characterization). Values in the range of 9–26 nm were found. On the other hand, the long period (*L_p_*), corresponding to the average length between amorphous domains separated by crystallites along the axis of the fiber, were obtained by synchrotron SAXS (see Appendix A for detailed characterization), achieving vales of the same order of magnitude (12–13 nm). The average length of the crystallites (*L_c_*) was found in the range of 3–4 nm, revealing characteristic length for the amorphous domains (*L_a_*) around 9 nm. These last parameters do not significantly change in the presence or in the absence of the copolymers.

The degree of crystallization of PCL in the mats could be calculated by DSC, comparing the enthalpy of melting for a pure PCL crystal and that of the different mats synthesized. No significant differences were found for the different fibers, showing values of crystallization of 50–56%, with the exception of PCL_394_/PEO_45_*-b-*PCL_11_ that showed 45%, as can be read in Table 1 (see Appendix A for detailed characterization).

#### 3.3.3. Thermal Properties

The glass transition temperature (*T_g_*), cold crystallization temperature (*T_c_*), and melting point (*T_m_*) of the fibers have been obtained by DMA and DSC (see Appendix A for detailed characterization). The storage modulus (DMA) of the studied samples decreases around −50 °C, associated to the *T_g_* of PCL. As long as temperature increases, a second pronounced decrease on the storage modulus is observed around 50 °C, which is assigned to the *T_m_* of PCL. The presence of the block copolymers did not produce significant changes in the *tan*
*δ* maxima, and thus in the *T_g_* and *T_m_* of the fibers, as can be read in Table 2. Similar values of *T_m_* were obtained by DSC. When measured before and after one heating and cooling cycle (*T_m_*^1^
_DSC_ and *T_m_*^2^
_DSC_, respectively), it is observed a decrease of around 3 °C. On the contrary, values of around 32 °C are found for *T_c_*, that remained stable after one heating and cooling cycle (*T_c_*^1^
_DSC_ and *T_c_*^2^
_DSC_, respectively). However, it is worth mentioning that the *T_c_* of the bulk PCL (as received before electrospinning processing) is 29 °C, indicating that the electrospinning process enhances nucleation during the formation of the fibers [70].

#### 3.3.4. Mechanical Properties

Tensile test of the fibrous materials (stress-strain curves) of PCL fibers showed Young modulus of PCL_394_ fibers of 8.3 ± 0.6 kPa, as can be seen in Figure 5. The presence of the block copolymers PEO_45_*-b-*PCL_11_ and PEO_148_*-b-*PCL_13_ in the fibers produces a slight decrease on the Young modulus, achieving values of 5.9 ± 1.2 and 6.3 ± 1.2 kPa, respectively, while it remains almost unchanged in the presence of PEO_230_*-b-*PCL_184_ (6.8 ± 1.7 kPa). The latter suggests that the loose of mechanical properties may be due to the smaller diameter of the fibers (see Figure 2). Elongation at the break has been performed up to 270% of deformation. In this range of deformation, PCL_394_ and PCL_394_/PEO_148_*-b-*PCL_13_ fibers do not break, meanwhile those prepared in the presence of PEO_45_*-b-*PCL_11_ and PEO_230_*-b-*PCL_184_ reach an elongation at break of 65% and 85%, respectively.

### 3.4. Biological Response of NSPC Cultured into the Fibrous Materials

6 DIV-neurospheres from the brain of SD rats were seeded in the fibrous materials in the absence of serum (growth factor) for 7 days. The phenotype of differentiated cells was analyzed by fluorescence microscopy through the expression of specific molecular markers and the analysis of cell morphology. These results are shown in the Figure 6. As a general view, all PCL based fibrous materials allowed neurospheres partially disassembling during the 7-day period. However, the degree of neurosphere disassembly increased in the presence of the PEO_m_-*b*-PCL_n_ copolymers. Interestingly, it was also possible to observe that the NSPC differentiate into β-III tubulin positive cells (neuroblasts/neurons) and GFAP positive cells (glioblasts/astrocytes) in the absence of any other stimuli such as growth factors. The presence of the PEO_m_-*b*-PCL_n_ copolymers in the materials revealed a faster acquisition of specific phenotype (molecular and/or morphology) features. However, astrocytes acquired typical molecular and morphological features, such as the expression of GFAP and short radiating processes that lined fiber mats (Figure 6), while the differentiation of neurons was not completed. Although they expressed a specific molecular marker (*β*-III tubulin), the neuritic extensions typical of neuronal morphology were not observed. This fact suggests that neuronal differentiation or neurogenesis may be delayed in relation to astrogenesis.

Special attention must be paid to the mats composed of PCL_394_/PEO_45_*-b-*PCL_11_ and PCL_394_/PEO_148_*-b-*PCL_13_ blends, those presenting a surface hydrophilic character. In these mats, astrocytes displayed earlier expression of molecular and morphological features, as can be seen in Figure 7. The differentiation response of astrocytes suggests that these cells are able to grow along the fibrous materials, using the microfibers as guidance. This fact is especially important. It is known that astrocytes are involved in many cellular processes, playing a crucial role in the central nervous system as well as in the early stage of brain development. Astrocytes act as guide for neurons [71], adding metabolic support and maintenance of the brain blood barrier [72]. Also, it is known that these cells are able to produce different component of the ECM such as laminin [73], thrombospondin-1 [74], and tenascin C [75], which could further help to guide the development of neurons. Thus, the presented scaffold could induce a posterior specific neuronal differentiation by means of directing astrocyte growing and networking.

The cellular and molecular mechanisms that guide the progression from a dividing NSC to a functional neuron are not fully understood. In the mammalian brain, neurogenesis is tightly regulated via both extrinsic environmental influences and intrinsic genetic factors [76,77,78]. Endogenous extrinsic factors include cell-to-cell interactions, niche-derived morphogens, growth factors and cytokines [79,80]. It is worth noting that, in the absence of some other stimuli such as growth factors, NSPC could differentiate in the surface of PCL fibers using the fibrous and/or porous arrangements as support for differentiation. Indeed, PCL scaffolds have been reported as supports for neural growing after addition of growth factors [49,81]. The findings shown here support the key role of ECM in the neurogenic niche. Surprisingly, this role has been far underestimated. In the mammalian brain the ECM and associated molecules may regulate signaling in the niche that acts as a substrate for anchoring cells and cell-cell contacts, committing so NSC to the acquisition of a neuronal or astrocyte phenotype. It is of great value the potential use of specific scaffoldings to uncovering the complex interactions within the neurogenic niche, as well as to promote restorative neurogenesis in brain disease or trauma.

## 4. Conclusions

In order to analyze the role of ECM in neurogenesis, we obtained micron-sized fiber mats by electrospinning based on PCL. The materials were composed of PCL_394_ and PEO_45_*-b-*PCL_11_, PEO_148_*-b-*PCL_13_, and PEO_230_*-b-*PCL_184_ block copolymers. The presence of PEO_m_*-b-*PCL_n_ block copolymers in the materials produces low significant changes in thermal, mechanical, and internal structure with respect to PCL_394_ mats. More significant changes were found in morphological features such as fiber diameter, decreasing in the presence of the copolymers, and surface properties. Surface topology show an increase in roughness as the PEO block increased in size. More interestingly, for both PCL_394_/PEO_45_*-b-*PCL_11_ and PCL_394_/PEO_148_*-b-*PCL_13_ mats, those containing copolymers bearing short blocks of PCL, the materials surface became hydrophilic and the materials wet in less than 1 s. The materials provided a suitable environment for the survival of NSPC. However, disassembling of seeded neurospheres is enhanced in the presence of the block copolymers. More importantly, the NSPC differentiate into neuroblasts/neurons and glioblasts/astrocytes in the absence of any other stimuli such as growth factors. After 7 days, astrocytes, mainly in the presence of PEO_45_*-b-*PCL_11_ and PEO_148_*-b-*PCL_13_, contained in the most hydrophilic mats, acquired typical morphological features such as short radiating processes that lined fiber mats, while typical neuritic extensions were not observed, suggesting that neuronal differentiation or neurogenesis is delayed in relation to astrogenesis. The presence of small amounts of PEO_m_*-b-*PCL_n_ block copolymers in PCL based mats allows controlling the surface hydrophilia of the materials, pointing at the modulation of the cellular response towards the disassembly and differentiation of NSPC.

## Figures and Tables

**Figure 1 polymers-11-01621-f001:**
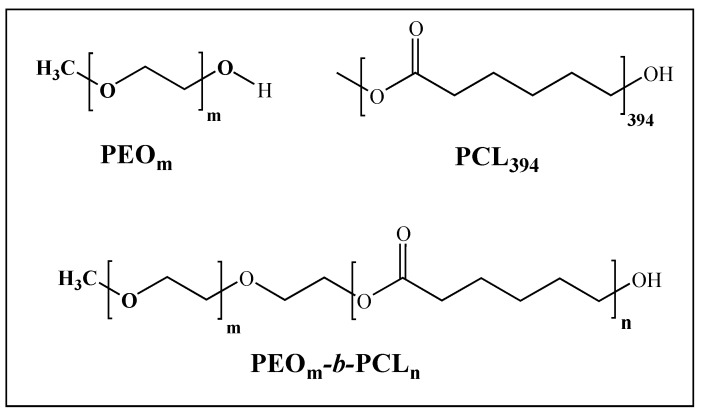
Molecular structures of polymers studied.

**Figure 2 polymers-11-01621-f002:**
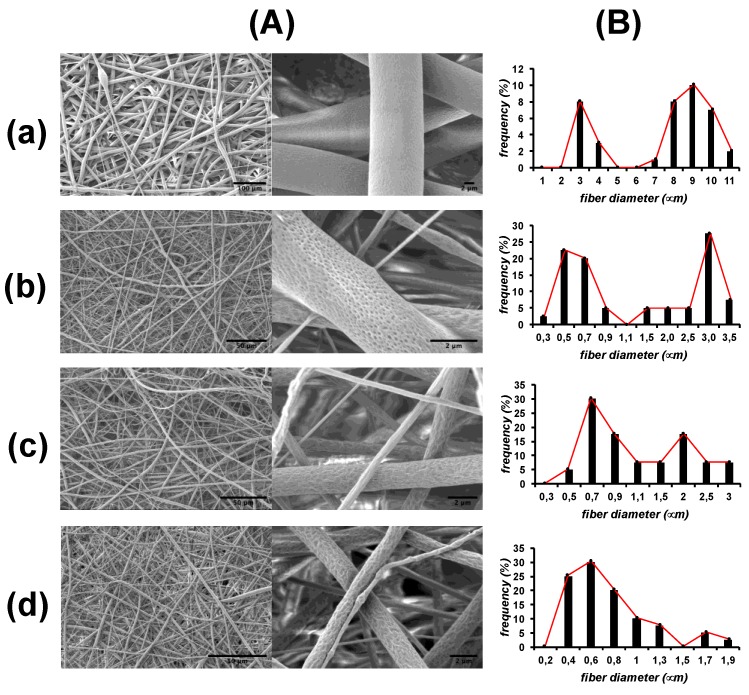
(**A**) SEM images and (**B**) fiber diameter distribution corresponding to fibers of poly(*ε*-caprolactone) (PCL_394_) prepared in the absence (**a**), and in the presence of 2.5 wt% of poly(ethylene oxide)_m_*-block-*poly(*ε*-caprolactone)_n_ (PEO_m_-b-PCL_n_) PEO_45_*-b-*PCL_11_, (**b**), PEO_148_*-b-*PCL_13_ (**c**), and PEO_230_*-b-*PCL_184_ (**d**).

**Figure 3 polymers-11-01621-f003:**
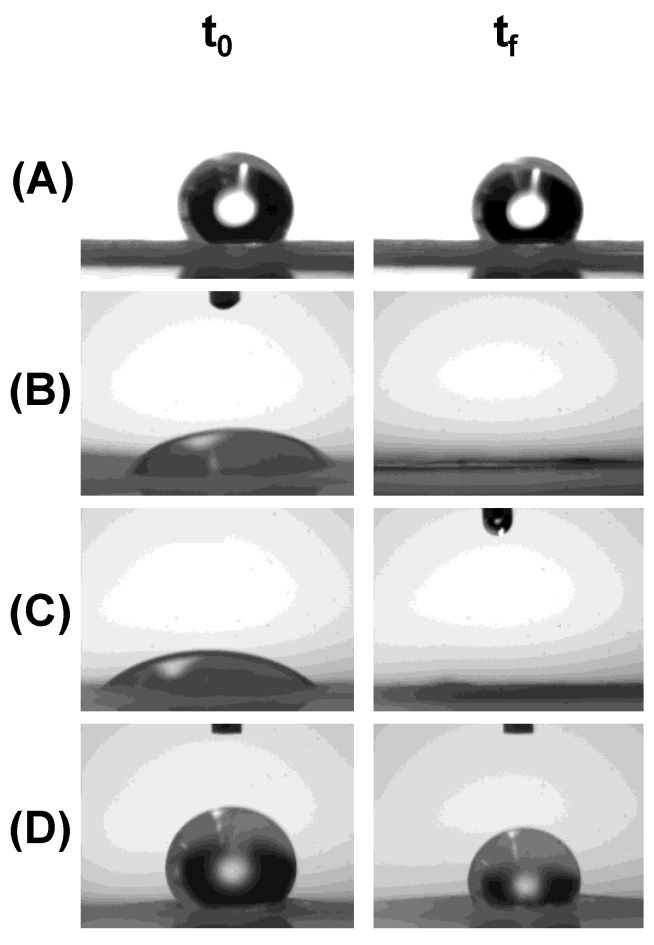
Wettability of PCL_394_ fibers prepared in the absence (**A**), and in the presence of 2.5 wt% of PEO_45_*-b-*PCL_11_ (**B**), PEO_148_*-b-*PCL_13_ (**C**), and PEO_230_*-b-*PCL_184_ (**D**), respectively.

**Figure 4 polymers-11-01621-f004:**
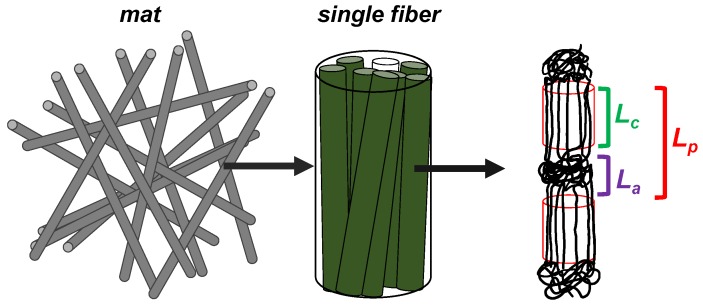
Schematic representation of crystalline and amorphous domains ordered in a PCL single fiber.

**Figure 5 polymers-11-01621-f005:**
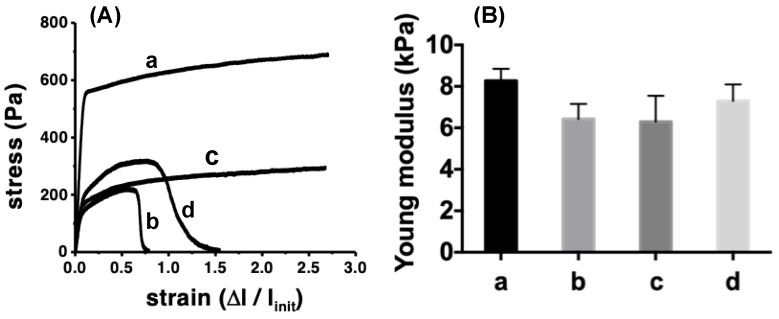
(**A**) Stress-strain plot and (**B**) calculated Young modulus for PCL fibers prepared using pristine PCL_394_ (**a**), and those prepared in the presence of PEO_45_*-b-*PCL_11_ (**b**), PEO_148_*-b-*PCL_13_ (**c**), and PEO_230_*-b-*PCL_184_ (**d**).

**Figure 6 polymers-11-01621-f006:**
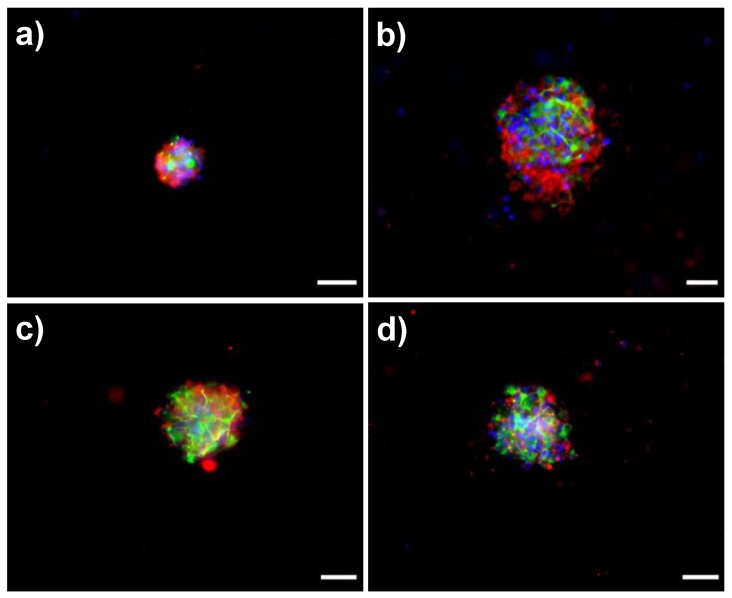
Epifluorescence microscopy images of differentiated neural stem progenitor cells (NSPC) into PCL fibrous mats: pristine PCL_394_ (**a**), and those prepared in the presence of PEO_45_*-b-*PCL_11_ (**b**), PEO_148_*-b-*PCL_13_ (**c**), and PEO_230_*-b-*PCL_184_ (**d**). Scale bar is 100 µm. Red marks are associated to β-III tubulin positive cells (neuroblasts/neurons); green marks to GFAP positive cells (glioblasts/astrocytes); and blue marks to cell nucleus (DAPI).

**Figure 7 polymers-11-01621-f007:**
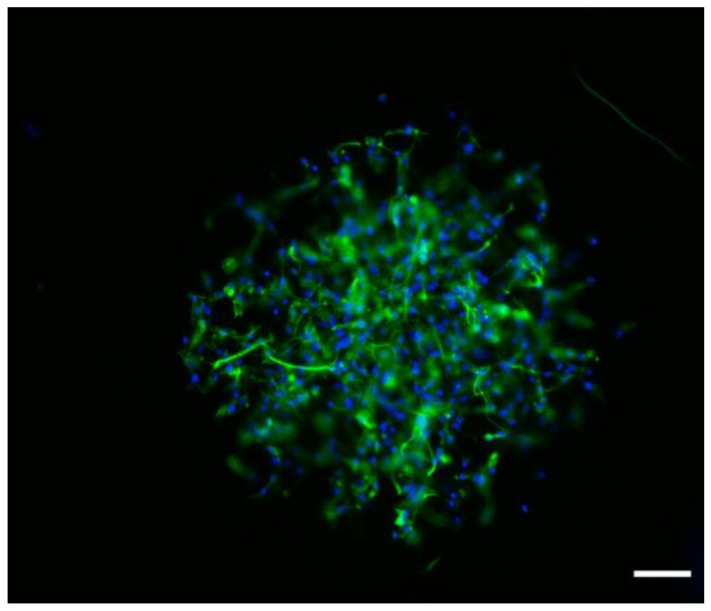
Interaction of astrocytes with fibers prepared with PCL_394_/PEO_148_-*b*-PCL_13_ blend and observed by epifluorescence microscopy. Scale bar is 150 µm. Green marks correspond GFAP positive cells (glioblasts/astrocytes); and blue marks to cell nucleus (DAPI).

**Table 1 polymers-11-01621-t001:** Characteristic lengths of crystalline and amorphous domains measured by X-ray diffraction (XRD) and small angle X-ray scattering (SAXS).

Mat Components	*L_(110)_* _(nm)_	*L_(200)_* _(nm)_	*L_p_* _(nm)_	*L_c_* _(nm)_	*L_a_* _(nm)_	*X_C_* _(%)_
PCL_394_	18	18	12.5	3.6	8.9	59.1
PCL_394_/PEO_45_*-b-*PCL_11_	19	21	12.2	3.0	9.2	45.1
PCL_394_/PEO_148_*-b-*PCL_13_	9	10	12.1	3.2	8.9	55.0
PCL_394_/PEO_230_*-b-*PCL_184_	26	23	12.1	3.4	8.7	55.2

**Table 2 polymers-11-01621-t002:** *T_g_* and *T_m_* of PCL materials measured by DMA (*n* = 3) and *T_m_* and *T_c_* studied by DSC before and after a heating/cooling cycle.

Sample	*T_g_*_DMA_ [°C]	*T_m_*_DMA_ [°C]	*T_m_*^1^_DSC_ [°C]	*T_c_*^1^_DSC_ [°C]	*T_m_*^2^_DSC_ [°C]	*T_c_*^2^_DSC_ [°C]
PCL_394_	−54.7 ± 5.3	65.2 ± 2.4	60.8	32.6	58.0	32.5
PCL_394_/PEO_45_-*b*-PCL_11_	−58.0 ± 4.2	61.0 ± 4.4	59.2	32.0	56.4	32.0
PCL_394_/PEO_148_-*b*-PCL_13_	−54.1 ± 5.6	64.8 ± 0.6	60.3	32.4	57.6	32.4
PCL_394_/PEO_230_-*b*-PCL_184_	−57.6 ± 3.9	59.1 ± 1.0	60.0	32.4	57.2	32.4

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
