# Peer review of "Fibrous Materials Made of Poly(ε-caprolactone)/Poly(ethylene oxide)-b-Poly(ε-caprolactone) Blends Support Neural Stem Cells Differentiation"

_polymers, 2019, doi:10.3390/polym11101621_

Round 1
Reviewer 1 Report
In this manuscript (polymers-600716) entitled “Fibrous Materials Made of Poly(ε-caprolactone) / Poly(ethyleneoxide)-b-Poly(ε-caprolactone) Blends Supports Neural Stem Cells Differentiation”, authors synthesized PEO-b-PCL block polymers with different molecular weights of PEO and PCL, and then applied electrospinning to produce a biomat from a blends of PCL and PEO-b-PCL. The morphology and properties of the mats were comprehensively characterized. Further the mats were used for the culture of Neural Stem Cells. The results indicated that the cells could be survived and differentiate into astrocytes and neurons. In general, it is an interesting work. However, there are still some issues to be addressed. This reviewer would suggest a minor revision before its acceptance.
Blend electrospinning is an efficient strategy to combine the advantages of different polymers. In introduction part, authors should demonstrated this point to show why authors used these two polymers for blend electrospinning. Please refer two of recent highly relevant references by blend electrospinning to produce high performance fibers: Polym. Chem. 9(20) (2018) 2685-2720; J. Mater. Sci. 53 (2018) 15096. The purity of solvents used in 2.1. should be clarified. How about the residual solvent in the fiber samples? Are there any effect of this residual solvent on the cell culture? 3.2 part should be divided into two sections, fiber fabrication and characterization. Is it possible to directly blend PEO and PCL for electrospinning to modify the wettability of mat? In section 3.3.3., authors provided the mechanical performance of mat. But the stress-strain curves ended at the strain of 0.5. Authors should perform complete measurement to show the curves. In addition, the unit for stress in the figure should be changed into standard unit. Figure 6 should separate the left and right figures a little.
Reviewer 2 Report
Taking into account all the several features (technical aspects, quality and presentation), the accuracy, scientific quality, scientific content and interpretation of the results are very good.
Technical
- The topic is appropriate for the journal.
The work has a very clear structure.
All the ideas are clearly and concisely expressed, and the concepts are understandable. The sections are well written in a way that is easy to read and understand.
The overall balance and structure of the paper is very good. Moreover, all the sections are necessary and properly written.
English language seems to be appropriate.
Quality
The paper deals with the design micron-sized fiber mats by blending poly(caprolactone) (PCL) with small amounts of block copolymers poly(ethylene oxide)m-block-poly(caprolactone)n (PEOm-b-PCLn) using electrospinning, reporting very interesting results. The authors start to discuss about tissue engineering and scaffolds. In the introduction they state: “A key topic in tissue engineering is to create a tridimensional scaffold or materials with suitable degradation rate, with the possibility to supply interconnected pores for cell-cell and cell-matrix interactions that bring cells together to form a tissue…”. As many works are reported in literature focusing on the design of 3D advanced scaffolds, I suggest to BRIEFLY introduce and cite some progress reporting analyses on different approaches in the design of polymeric and composite scaffolds for tissue engineering using advanced technologies such as additive manufacturing techniques (i.e., FDM) (i.e., “Three-dimensional printed bone scaffolds: The role of nano/micro-hydroxyapatite particles on the adhesion and differentiation of human mesenchymal stem cells”. Proc Inst Mech Eng Part H J Eng Med 2017; 231:555–564.…). Then, the authours should continue to stress their approach and study related to the design micron-sized fiber mats by blending poly(caprolactone) (PCL) with small amounts of block copolymers poly(ethylene oxide)m-block-poly(caprolactone)n (PEOm-b-PCLn) using electrospinning. This should improve the quality of the paper reporting a wide range of progresses/approaches in the field of bone tissue engineering and should help the different kinds of readers to better understand the value of their work.
The approach is interesting.
The introduction should be improved
The List of references should be improved.
It seems that the paper does not contain repetitions.
The length of the work is appropriate and consistent with its scientific content.
Presentation
The quality of some figures should be improved. The title is adequate and appropriate for the content of the article. The abstract contains information of the article. Figures and captions are essential and clearly reported.
